

# Balance performance, falls-efficacy and social participation in patients with type 2 diabetes mellitus with and without vestibular dysfunction

Dwi Rosella Komalasari[1], Mantana Vongsirinavarat[2], Vimonwan Hiengkaew[2] and Nantinee Nualnim[2]

[1] Physiotherapy, Muhammadiyah University of Surakarta, Surakarta, Central Java, Indonesia
[2] Physical Therapy, Mahidol University, Salaya, Phuttamonthon, Nakhon Pathom, Thailand

## ABSTRACT

**Background**. The performance of balance is an important factor to perform activities. The complications of type 2 diabetes mellitus (T2DM), especially vestibular dysfunction (VD), could decrease balance performance and falls-efficacy (FE) which consequently impacts social participation and quality of life (QoL).

**Purpose**. This study aimed to compare balance performance, FE, social participation and QoL between individuals with T2DM with and without VD.

**Methods**. The participants comprised 161 T2DM with VD and 161 without VD. Three clinical tests used for confirming VD included the Head Impulse Test (HIT), the Dix Hallpike Test (DHT) and the Supine Roll Test (SRT). The scores of static and dynamic balances, FE, social participation and QoL were compared between groups.

**Results**. The balance performance, FE, social participation and QoL were lower in the group with VD. The number of patients who had severe social restriction was higher in T2DM with VD than without VD (58.4% *vs* 48.4%). Moreover, all domains of QoL (physical, psychological, social relationships and environmental) were lower in T2DM with VD than without VD.

**Conclusion**. The presence of VD in T2DM patients was associated with decreased physical balance performances and increased social and QoL disengagement. Comprehensive management related to balance and FE, as well as the monitoring to support social participation and QoL, should be emphasized in patients with T2DM with VD.

## INTRODUCTION

Inputs from three systems, including visual, proprioception and vestibular, play important roles in postural control, especially when performing dynamic tasks. The integration of these sensations regulates appropriate timed and scaled outputs to preserve the postural control (*Mancini & Horak, 2010*; *Sibley et al., 2015*). A greater number of impaired sensory systems is presented with different levels of difficulties related to balance and falls (*Kraiwong et al., 2019*). The impairment of sensory systems and lower balance performance in patients

Corresponding author
Dwi Rosella Komalasari,
drks133@ums.ac.id

with T2DM are associated with microvascular complications. Neuropathy (DPN) is the most common complication that diminished the function of proprioception, presenting in 60–70% of chronic T2DM (*Centers for Disease Control and Prevention., 2011*; *Purwanti & Novitasari, 2021*). DPN alters movement as a result of the diminished perceptions of touch, temperature, and pain, and reduced proprioceptive inputs leading to difficulty in ambulation and falling (*Brod et al., 2015*). The DPN can develop into the foot ulcers without good management that can interfere the gait and increase fall (*Rosyid, 2017*). Another common complication of T2DM, the diabetic retinopathy (DR) which impacts the reduction in visual system, had also been reported to increase with age and brought a 25 times higher risk of losing vision compared with people without DM (*Jenkins et al., 2015*). These two common microvascular complications largely contributed to the elevated postural sway (*Brod et al., 2015*; *Schwartz et al., 2008*).

Along with the visual and somatosensory functions, the vestibular system also plays an important role in controlling balance especially when the head moved. Vestibular dysfunction (VD) has been recognized as an independent factor of falling. Diminished sensitivity of the vestibular system can alter perception regarding motion, equilibrium, and spatial orientation which is necessary to maintain stability (*Goble et al., 2019*). A survey in the US found a VD prevalence of 70% among participants with DM aged 40 years or older (*Agrawal et al., 2009*). The risk of falling increases two-fold in DM patients with VD who had long duration of DM (≥5 years) compared with the group without VD (*Agrawal et al., 2010*). Moreover, dizziness which is the most common symptom of VD was reported in 20–30% of the general population. Participants with symptomatic VD had dizziness symptoms and had 12 times increasing of falling (*Agrawal et al., 2009*).

Falls-efficacy (FE) is the self-perceived rating to perform activities of daily living without falling (*Tinetti, Richman & Powell, 1990*). This factor has been proposed to be a critical indicator of diabetes self-care and a risk factor for falls. Reduction of self-efficacy becomes self-blame, worse self-complaint, and criticisms from others that might further affect the quality of life (QoL). The low self-efficacy were prevalent among older adults, affecting 33 to 46% of older adult non-fallers and up to 85% of older adult fallers (*Kumar et al., 2014*).

The World Health Organization has declared social participation as a substantial component of active aging (*WHO, 2002*). Social participation is defined as the involvement in activities and interactions with other people in a social or community frame. Persons with DM reported higher probability of a lack of social involvement, but the contribution of disability related with DM to the social life has not been fully explored (*Brinkhues et al., 2017*; *Weidt et al., 2014*).

The aging process is a factor deteriorating the body function in elderly including balance control. A previous study reported that people with DM had poorer balance than without DM. Diminished balance performance is occurred in DM, even without sensory impairment (*Kraiwong et al., 2019*). The consequences of DPN and DR associated with T2DM on gait instability and falls are well documented. However, the information about VD effects is rather poorly recognized (*Agrawal et al., 2010*; *D'Silva et al., 2016*). The VD symptoms of dizziness and imbalance were considered to be the main predictors of falling in the elderly and might also decrease balance confidence. However, few studies had evaluated the FE
associated with diminished balance and increased fear of falling (FoF) in patients with DM. A recent study reported that T2DM patients with VD had severe restrictions in social participation and also had movement difficulty due to the dizziness feeling (*Komalasari et al., 2022*). Another study showed that among 34 activities in the Vestibular Activities Participation (VAP) questionnaire, 10 activities were difficult to be carried out by VD patients (*Alghwiri, Alghadir & Whitney, 2013*). The inference of having VD on the factors associated with falls including balance performance, fall-efficacy, social participation and quality of life among individuals with T2DM needs more exploration. Therefore, this study aimed to compare balance performance, falls-efficacy, social participation and quality of life between individuals with T2DM with and without vestibular dysfunction.

## MATERIALS AND METHODS

This study was a cross-sectional, comparative research. It was conducted after the approval from the ethics committee of Mahidol University Institutional Review Board (MU-CIRB 2020/098.2004) and the Medical Faculty, Universitas Muhammadiyah Surakarta, Indonesia (No. 3009/B.1/KEPK-FKUMS/VII/2020).

### Participants

The participants were individuals with DM who lived in the community. Comparable with the previous study, the inclusion criteria were an age older than 40 years old and being diagnosed with T2DM at least 5 years. The participants were excluded if they had (1) amputation of the lower extremity, (2) blindness or a visual impairment, (3) foot ulcers, (4) orthopaedic or surgical problems affecting gait, (5) macrovascular condition such as cardiovascular disease, (6) history of brain injury, (7) central nervous system lesions (stroke, Parkinson, cerebellar ataxia), (8) unstable medical condition, (9) using assisted devices for standing and walking, (10) inability to stand or walk independently and (11) inability to follow verbal commands.

Since this report is a part of main prevalence study, the sample size was calculated based on the level of confidence (Z) of 1.96, the expected prevalence (P) of 22%, and precision (d) of 0.05. The prevalence was based on the report by *Agrawal et al. (2010)* which conducted the survey in the population of DM. The formula used sample size for cross sectional study, that was $n = Z_{1-\alpha/2^2} xPx(1-P)/d^2$. We planned to add 25% of subjects to prevent attrition because the testing session was quite length and took two days to finish all testings. Therefore, the planned sample size collection was 165 participants per group. However, there was some incomplete data of participants in each groups, so finally the study included 161 participants. All participants were informed about the research process and signed the informed consent.

### Procedure

Personal and clinical characteristics were recorded. The number of falls was reported using the previous twelve months timeframe. The glycemic level was tested on the physical examination day from the fasting plasma glucose (FGL) of the intravenous blood. The glycemic level was tested on the physical examination day from the fasting plasma

glucose (FGL) of the intravenous blood. The blood pressure measured by automatic sphygmomanometer and devided into hypertension (>140/90) mmHg and normal blood pressure (≤140/90) mmHg (*WHO, 2013*). The body mass index (BMI) was a person's weight in kilograms devided by the square of height in meters. BMI was classified into underweight (<18.5), normal weight (18.5–24.9), overweight BMI (≥25.0), and obesity (≥30.0) (*WHO, 2023*).

The cognitive function was examined using the MoCA Indonesian version (MoCA-Ina) (*Husein et al., 2010*). The test-retest reliability of MoCA-Ina was reported to be acceptable (ICC 0.820) (*Husein et al., 2010*). Its total score was 30; the score 18–25 was categorized as having a mild cognitive impairment, 10–17 was moderate cognitive impairment and less than 10 was severe cognitive impairment (*Nasreddine et al., 2005*).

The vestibular dysfunction was confirmed by three tests including the Head Impulse Test (HIT), Dix Hallpike Test (DHT), and Supine Roll Test (SRT). The HIT assesses the functions of the horizontal semicircular canal (HSCC) and superior vestibular nerve (*Schubert et al., 2004*). It had sensitivity and specificity of 71% and 82% in patients with unilateral vestibular hypofunction, as well as 84% and 82% respectively in bilateral vestibular hypofunction (*Schubert et al., 2004*). The posterior benign paroxysmal positional vertigo (BPPV) was assessed by the DHT (*Noda et al., 2011*). The SRT was used to identify the horizontal BPPV (*Lim et al., 2013*).

The DPN was screened by physical examination according to the Michigan Neuropathy Screening Instrument (MNSI). This tool has sensitivity and specificity of 61% and 79%, respectively, for examining clinical neuropathy with a cut-off score of 2.5 (*Herman et al., 2012*). Pain at the lower extremity associated with DPN was also assessed by the Numeric Rating Scale (NRS). The overall intensity within that week was rated. The type of pain could be burning, tingling, electric, sharp, shooting and lancinating. The range of the NRS ranges from zero to ten, with zero indicating "no pain" and 10 indicating "the worst pain imaginable" (*Kugbey, Oppong Asante & Adulai, 2017*; *Sadosky, Hopper & Parsons, 2014*). The visual acuity was identified by a letter chart. The participants were asked to sit in front of the chart and read the letters on the chart from the biggest to the smallest letters that they could read. They could use their proper glasses during the test. The score was obtained using a visual acuity conversation, with a score ≥ 75 considered normal visual acuity (*Colenbrander, 2010*). In addition, the number of impaired sensory system was determined from the results of vestibular dysfunction, neuropathy and visual tests.

### Balance performance

The static balance was measured by the Modified Clinical Test of Sensory Interaction of Balance (mCTSIB). This test is highly sensitive (88–91%) and moderately specific (50–57%) when using the Sensory Organization Test as a reference standard (*Wrisley & Whitney, 2004*). The participants were asked to stand with feet together for 30 s under four testing conditions: (1) eyes open and firm surface, (2) eyes closed and firm surface, (3) eyes open and foam surface, and (4) eyes closed and foam surface. The foam used in this study had medium density with a size of 24 inches in width and length and 4 inches in height (SunMate; Dynamic System Inc., Leicester, NC, USA). The participants performed

three trials of each condition, and the average time was used to represent the condition. If the time was less than 30 s, the performance of that condition was considered failed. The sum of the average time of four conditions was used to represent the total mCTSIB score (mCTSIB-Tol).

The dynamic balance was measured by the Timed Up and Go test (TUG) and the Functional Gait Assessment (FGA). The TUG test is a test used to measure balance during mobility that patients perform walking for 3 m and back to sit again (*Podsiadlo & Richardson, 1991*). This test could predict fall with the sensitivity and specificity of 80% and 56% respectively (*Whitney et al., 2004*). The FGA is a test assessing postural stability during walking and performing multiple motor tasks. For peripheral vestibular patients, FGA had excellent intra-rater reliability ICC = 0.94 (95% CI [0.85–0.97]), agreement was moderate to high for all items and weighted kappa was good to very good. The inter-rater was reported having ICC 0.73 (95% CI [0.49–0.86]), agreement between raters was 0.45−0.91 and weighted kappa was 0.27 to 0.87 (*Nilsagård et al., 2014*). The correlation between FGA and TUG was 0.84 (*Wrisley & Kumar, 2010*).

The functional lower extremity (LE) muscle strength was measured by the Five Time Sit to Stand Test (5XSST). A 5XSST of 13 s indicates balance dysfunction in individuals with balance disorder or VD. Also, it represents moderate sensitivity and specificity of 66% and 67% respectively (*Whitney et al., 2005*).

## Falls-efficacy

Falls-efficacy is defined as the confidence to carry out some particular activities without feeling of fall (*Whitney, Hudak & Marchetti, 1999*). The Activities-specific Balance Confidence Scale (ABC) with 16 items reflects a deteriorating physical ability to maintain balance in activities and is commonly used to measure balance confidence in people with balance and vestibular disorders (*Powell & Myers, 1995*). There are 16 items with a 10 point ordinal scale. The score ranges from 0 to 100, with categorized 0 as no confidence to 100 indicating complete confidence with 16-item questions. The ABC scale had an excellent Cronbach's Alpha of 0.95 applied to VD patients with peripheral VD (*Karapolat et al., 2010*). *Friscia et al. (2014)* adopted the cut score of ≤ 67 to have a greater responsiveness significantly to indicate increase risk of falls in individuals with vertigo, dizziness and unstable.

## Social participation

Measuring the social life shows the QoL and both the indicator of health, well-being and social behavior. People with VD have some problems inhibit their activities. Measuring the level of social participation is significant to design the management, development and evaluation to reduce the stigma disability of T2DM patient with VD. Social participation was assessed by the Indonesian version of the Life Space Assessment (LSA-Ina) and the Indonesian version of the Participation Scale Short Simplified (PSSS-Ina). The LSA is a self-report measure, that requires respondents to quantify how far and how often they have mobilized from home to beyond town or region, with or without assistance during the last 4 weeks (*Stalvey et al., 1999*). This tool was applicable to use in a community setting and

had satisfactory construct validity and was sensitive to change over a short time frame. It consists of five questions about the level of mobility. Every question must be answered by "yes" or "no"; if the patients answered "yes", they continued to explain the frequencies of the mobility. However, if the answer is "no", it means the score is zero. The total scores ranged from 0 (totally bed-bound) to 120 (moved out of town every day without assistance) (*Stalvey et al., 1999*). The PSSS was also used to capture the quality of social participation in a different perspective. This scale has been done for cultural adaptation into the Indonesian language (PSSS-Ina). It was reported to have an excellent ICC of 0.93 for T2DM with VD in Indonesia. The score of 31 identified the severe participation restriction in T2DM with VD (*Komalasari et al., 2022*).

The QoL was monitored by the WHOQoL-Bref tool. This instrument has been proposed to reflect the level of QoL for any stage. The WHOQoL-Bref Indonesian version had excellent discriminant validity, construct validity and good internal consistency in the elderly (*Oktavianus et al., 2007*).

### Data analysis

The statistical analysis was performed using the statistical software package SPSS for Windows version 25 (SPSS Inc., Chicago, IL, USA). The Kolmogorov–Smirnov Goodness of Fit test was used to test the distribution of data. Demographic data and physical examination information were compared between groups. The Chi-square test was used, to examine the difference in categorical variables between groups, and the $t$-test with two-tailed was used in continuous variables.

## RESULTS

The participants consisted of 161 T2DM with VD and 161 T2DM without VD. As presented in Table 1, there were no significant differences between the groups for the personal characteristics.

The numbers of female participants were greater in both groups. The participants in both groups had the same average age and around half of them had hypertension. No differences about hypertension and normal tension between two groups. Likewise, around three-fourths of the participants in both groups suffered from T2DM between 5–10 years. Regarding the cut level of fasting period by the World Health Organization, most of them also had high fasting glycemic levels (>125 mg/dl) (*WHO, 1999*), especially in the group with VD which 72% were presented with high FGL levels. More participants had normal BMI; nevertheless, almost 50% of the participants were overweight in both groups. Approximately one-fifth of participants in both groups experienced a fall once in the past year.

The results of clinical examination are summarized in Table 2. There were no significant differences for pain, DPN, and cognitive function between T2DM with VD and without VD groups. The lower leg pain was on average five of 10 measured by NRS for both groups. The score and proportion of participants with DPN were not different between groups. The average score of visual acuity was different but there was no difference of the percentages of participants with low visual acuity between groups.

**Table 1 The characteristics of the participants.**

| Participants characteristic | T2DM with VD ($n = 161$) | | T2DM without VD ($n = 161$) | | p-value |
|---|---|---|---|---|---|
| | Mean ± SD | N (%) | Mean ± SD | N (%) | |
| Age (years) | 61.6 ± 6.7 | | 61.7 ± 6.1 | | 0.85 |
| Sex | | | | | |
| (1) Male | | 72 (44.7) | | 68 (42.2) | 0.10 |
| (2) Female | | 89 (55.3) | | 93 (57.8) | |
| Blood pressure | | | | | |
| (1) Systolic | 143.6 ± 20.7 | | 142.6 ± 22.8 | | 0.67 |
| (2) Diastolic | 87.3 ± 10 | | 86.3 ± 9.9 | | 0.38 |
| (1) Yes (≥140/90 mmHg) | | 83 (51.6) | | 79 (49.1) | 0.27 |
| (2) No (<140/90 mmHg) | | 78 (48.4) | | 82 (50.9) | |
| BMI | 24.7 ± 1.9 | | 24.8 ± 2.1 | | 0.64 |
| (1) Normal | | 87 (54) | | 87 (54) | 0.07 |
| (2) Overweight | | 67 (41.6) | | 72 (44.7) | |
| (3) Obesity | | 7 (4.3) | | 2 (1.2) | |
| FGL (mg/dl) | 157.2 ± 47.7 | | 152.9 ± 49 | | 0.66 |
| (1) Normal | | 45 (28) | | 54 (33.5) | 0.54 |
| (3) Hyperglycemia | | 116 (72) | | 107 (66.5) | |
| Duration of DM (years) | 8.8 ± 3.3 | | 10.2 ± 8.6 | | 0.43 |
| (1) 5–10 | | 122 (75.8) | | 123 (76.4) | 0.16 |
| (2) ≥ 11 | | 39 (24.2) | | 38 (23.6) | |
| Number of falling | 0.3 ± 0.6 | | 0.3 ± 0.5 | | 0.61 |
| (1) No | | 122 (75.8) | | 124 (77) | 0.73 |
| (2) Once | | 29 (18.1) | | 30 (18.6) | |
| (3) 2 or more | | 10 (6.2) | | 7 (4.3) | |

Notes.
  VD, vestibular dysfunction; BMI, body mass index; FGL, fasting glycemic level; DM, diabetes mellitus.

The time of 5XSST was significantly different ($p < 0.001$) between groups. The average score of Moca-Ina and the proportion of ones with cognitive impairment were not different between groups. Both groups had persons mostly with mild and some with moderate cognitive impairment.

As shown in Table 3, the static balance measured by total scores of mCTSIB test was significantly different between two groups ($p < 0.001$). The group with VD had lower static balance performance compared to those without VD. Likewise, conditions 2 ($p = 0.02$), 3 and 4 ($p < 0.001$) of mCTSIB were significantly different between groups. However, the time used in condition 1 of mCTSIB was not significantly different between groups ($p = 0.357$). For dynamic balance, the total scores of TUG ($p = 0.001$) and FGA ($p = 0.01$) were significantly different between groups. The total scores of FE measured by ABC-16-Ina were significantly different between two groups ($p < 0.001$). FE in T2DM without VD group was higher than in T2DM with the VD group.

There were significant differences for PSSS and LSA among groups, and the T2DM with VD group had lower social quality than T2DM without VD measured by LSA and

**Table 2  The clinical examination results.**

| Tests | T2DM with VD (n = 161) | | T2DM without VD (n = 161) | | p-value |
|---|---|---|---|---|---|
| | Mean ± SD | N (%) | Mean ± SD | N (%) | |
| Neuropathic pain (NRS) | 5.7 ± 1.2 | | 5.7 ± 1.3 | | 0.89 |
| DPN | 1.9 ± 1.3 | | 2.0 ± 1.3 | | 0.68 |
| (1) Positive | | 85 (52.8) | | 66 (41) | 0.08 |
| (2) Negative | | 76 (47.2) | | 95 (59) | |
| Visual acuity | 76.8 ± 6.1 | | 75.3 ± 6.4 | | 0.03* |
| (1) Normal (≥75 points/letters) | | 40 (24.8) | | 31 (19.3) | 0.34 |
| (2) Low (<75 points/letters) | | 121 (75.2) | | 130 (80.7) | |
| Number of impaired sensory system(s) | | | | | |
| 0 | | – | | 24 (15%) | 0.25 |
| 1 | | 129 (80.1%) | | 49 (30.6%) | |
| 2 | | 27 (16.8%) | | 87 (54%) | |
| 3 | | 5 (3.1%) | | – | |
| 5XSST (secs) | 16.5 ± 3.3 | | 14.4 ± 2.6 | | <0.001* |
| Moca-Ina | 21.8 ± 3.3 | | 21.7 ± 3.4 | | 0.81 |
| (1) Mild impairment (score 18–25) | | 113 (70.2%) | | 111 (68.9%) | 0.40 |
| (2) Moderate impairment (score 10–17) | | 18 (11.2%) | | 23 (14.3%) | |
| (3) Severe impairment (Score <10) | | – | | – | |

Notes.
*Significantly different between groups.
T2DM, type 2 diabetes mellitus; VD, vestibular dysfunction; NRS, numeric rating scale; DPN, Diabetic Peripheral Neuropathy; 5XSST, Five Time Sit to Stand Test; Moca-Ina, Montreal Cognitive Assessment-Indonesian version.

PSSS. The number of participants with severe restriction was higher in T2DM with VD group than without VD. The total scores and all domains of WHOQoL were also different between groups ($p < 0.001$). The T2DM with VD had lower QoL compared with T2DM without VD.

Figure 1 describes mean rank and shows that the group without VD had a higher score than group with VD. But for condition 1 of mCTSIB it is shown that the violin plot has no any differences for mean, median score and inter quartile range (IQR). There were 25% the data value in condition 1 show had the same score between group 1 and 2. The upper quartile explained condition 1 and 2 had the same score in both groups. However, the variability of the data for conditions 2 and 4 were higher in the group with VD. The variability of the data of mCTSIB total and condition 3 presented higher in the group without VD. Furthermore, Fig. 2 explains almost all the data had variability distribution, exception the TUG (both groups) and social participation had a higher probability density function in the group with VD. The variability of the data was higher in the group with VD for FGA, FE, LSA and WHOQoL, while, TUG and PSSS were higher in the group without

**Table 3  The comparisons of balance performance, falls-efficacy, and social participation between T2DM with the VD group and T2DM without the VD group.**

| Tests | T2DM with VD | | T2DM without VD | | *p*-value |
|---|---|---|---|---|---|
| | Mean ± SD | N (%) | Mean ± SD | N (%) | |
| mCTSIB total | 84.1 ± 13.7 | | 91.1 ± 13.6 | | <0.001* |
|    Condition 1 | 28.8 ± 3.3 | | 28.7 ± 3.2 | | 0.35 |
|    Condition 2 | 24.7 ± 6.4 | | 26.1 ± 5.5 | | 0.02* |
|    Condition 3 | 18.6 ± 6.5 | | 21.8 ± 6.5 | | <0.001* |
|    Condition 4 | 12.3 ± 6.5 | | 14.5 ± 3.3 | | <0.001* |
| TUG | 15.1 ± 2.5 | | 14.0 ± 2.4 | | 0.001* |
| FGA | 21.2 ± 3.3 | | 22 ± 2.9 | | 0.01* |
| ABC-16-Ina | 64.6 ± 4.9 | | 65.6 ± 5.3 | | <0.001* |
| PSSS-Ina | 32.3 ± 5.5 | | 30.9 ± 5.5 | | 0.03* |
| (1) Severe restriction (Score 31–50) | | 94 (58.4) | | 78 (48.4) | |
| (2) Moderate restriction (Score 14–30) | | 67 (41.6) | | 83 (51.6) | <0.001* |
| LSA-Ina | 60.1 ± 14.8 | | 68.1 ± 9.3 | | 0.002* |
| WHOQoL-Bref-Ina | 65.3 ± 6.2 | | 72 ± 7.5 | | <0.001* |
|    Physical domain | 18.8 ± 1.8 | | 21.4 ± 2.5 | | <0.001* |
|    Psychological domain | 17.3 ± 1.6 | | 18.6 ± 2.4 | | <0.001* |
|    Social Domain | 9.5 ± 1.3 | | 10.7 ± 2.5 | | <0.001* |
|    Environmental domain | 60.1 ± 14.8 | | 21.3 ± 5.0 | | <0.001* |

**Notes.**
*Significantly different between groups.
mCTSIB, Modified Clinical Test of Sensory Interaction of Balance; TUG, Timed Up and Go test; FGA, Functional Gait Assessment; ABC-16-Ina, Activities-specific Balance Confidence Scale-16-Indonesian version; PSSS-Ina, Participation Scale Short Simplified-Indonesian version; LSA-Ina, Life Space Assessment-Indonesian version; WHOQoL-Bref-Ina, World Health Organization Quality of Life-Bref-Indonesian version.

VD. Interestingly, the data value about 25% also seemed for environmental domain (Fig. 3). No difference for inter quartile range (IQR) psychological and environmental domain for both groups. But, in the group without VD had higher variability of data for physical and social domain.

# DISCUSSION

This study showed the differences of balance performance, falls-efficacy, social participation and quality of life between individuals with T2DM with and without VD. Overall, the T2DM participants with VD had lower performance of balance and falls-efficacy, as well as reported poorer social participations and quality of life.

In this study, mostly the participation had duration of DM of 5–10 years. *Komalasari et al. (2022)* also reported the same result but different year and area (*Komalasari & Arif, 2023*). Both static and dynamic balance performances were evaluated by mCTSIB, TUG and FGA. All balance test results indicated lower ability in the group with VD. The mCTSIB has been used to detect balance impairment by specifically indicating sensory contributing for balance control (*Wrisley & Whitney, 2004*). There was no difference between groups

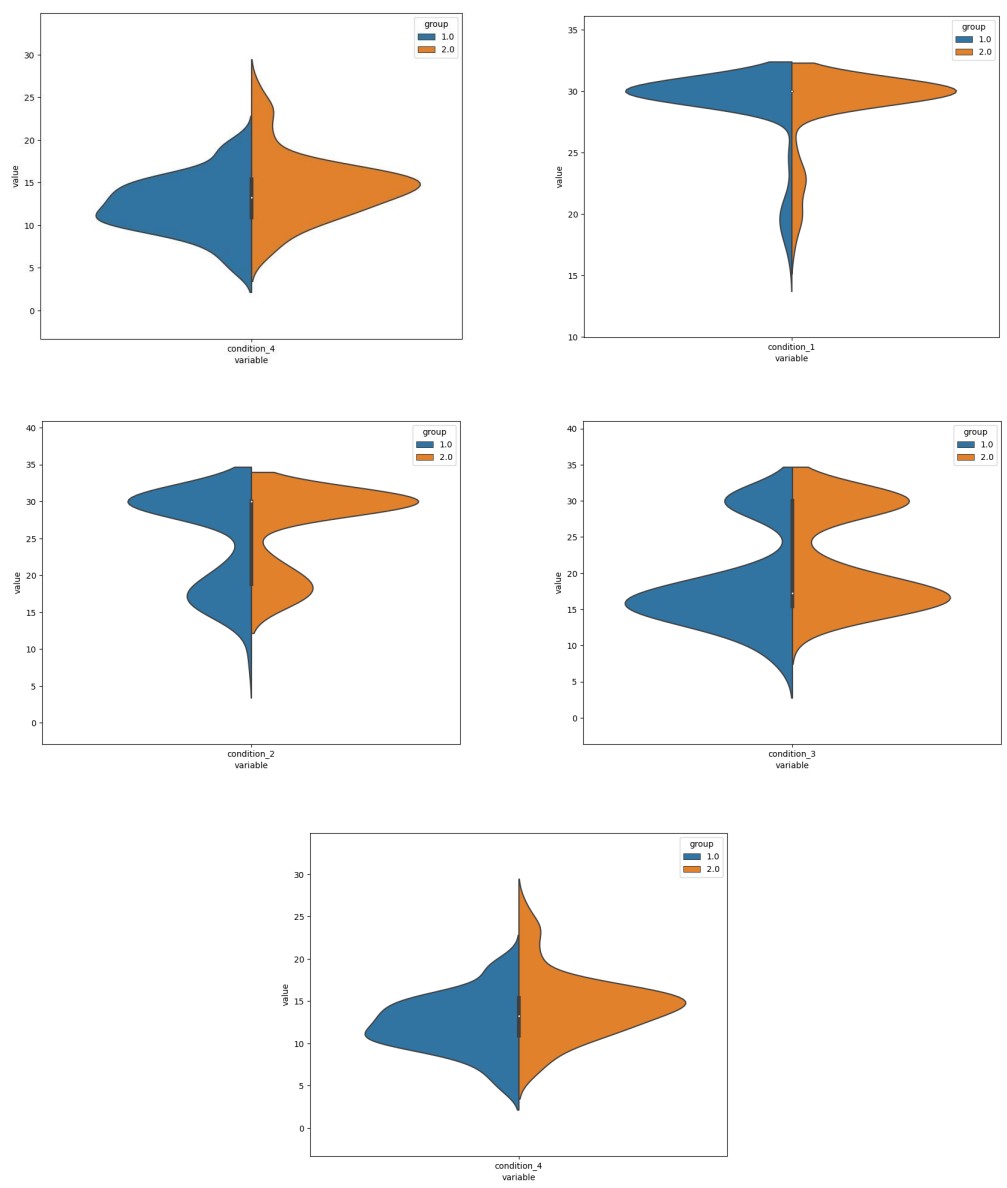

**Figure 1  The violin plots of mCTSIB tests.**

for condition 1 but significant differences for total score as well as conditions 2, 3 and 4 in this study. This results also confirmed that the limited function of the vestibular system in patients with VD affected the balance performance. Condition 1 is a standard test where the person uses all three sensory systems available to maintain balance, and therefore the smallest amount of postural sway is expected (*Goble et al., 2019*). In condition 2, in which the participants stand on a firm surface with eyes closed, *i.e.,* eliminating visual feedback, the proprioceptive and vestibular systems were delegated to detect the body position. This condition is postulated to measure the contribution of proprioception in balance

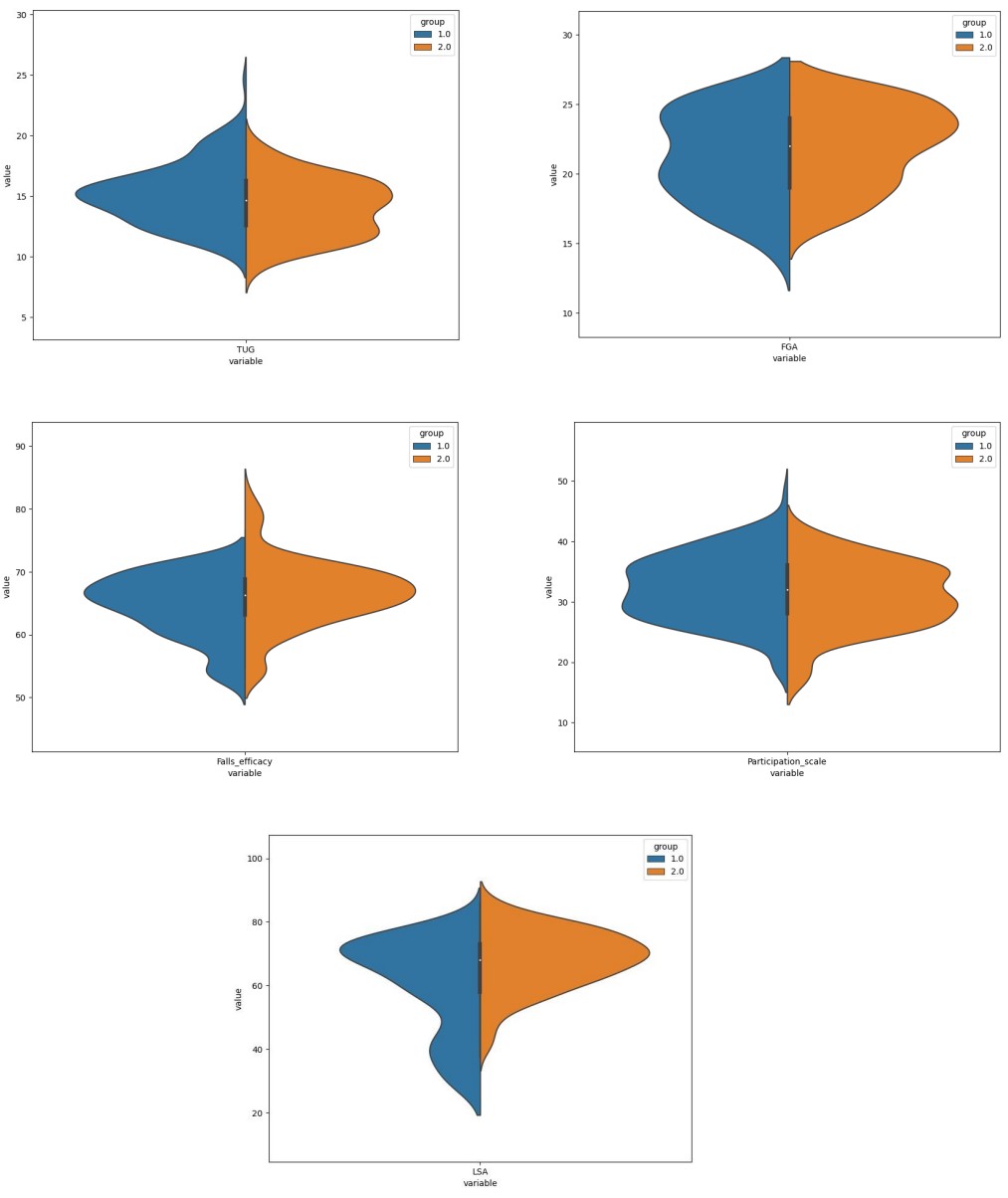

**Figure 2   The violin plots of TUG, FGA, falls-efficacy, participation scale and LSA.**

mostly (*Nashner, Black & Wall 3rd, 1982*). In this study, the patients with VD had lower performance under condition 2 compared with the ones without VD. It should be noted that more than half of participants in both groups had DPN which might also be associated with the proprioceptive impairment. The violin plot (picture 1) which describes condition 2 also has the same median range and upper quartile in both groups. This result confirmed the VD patient could still stand on foam with eyes opened. The role of the visual system is very useful for stabilizing body position despite having limitations in the vestibular system. Gaze stabilization is the ability of individuals to focus their eyes when their head
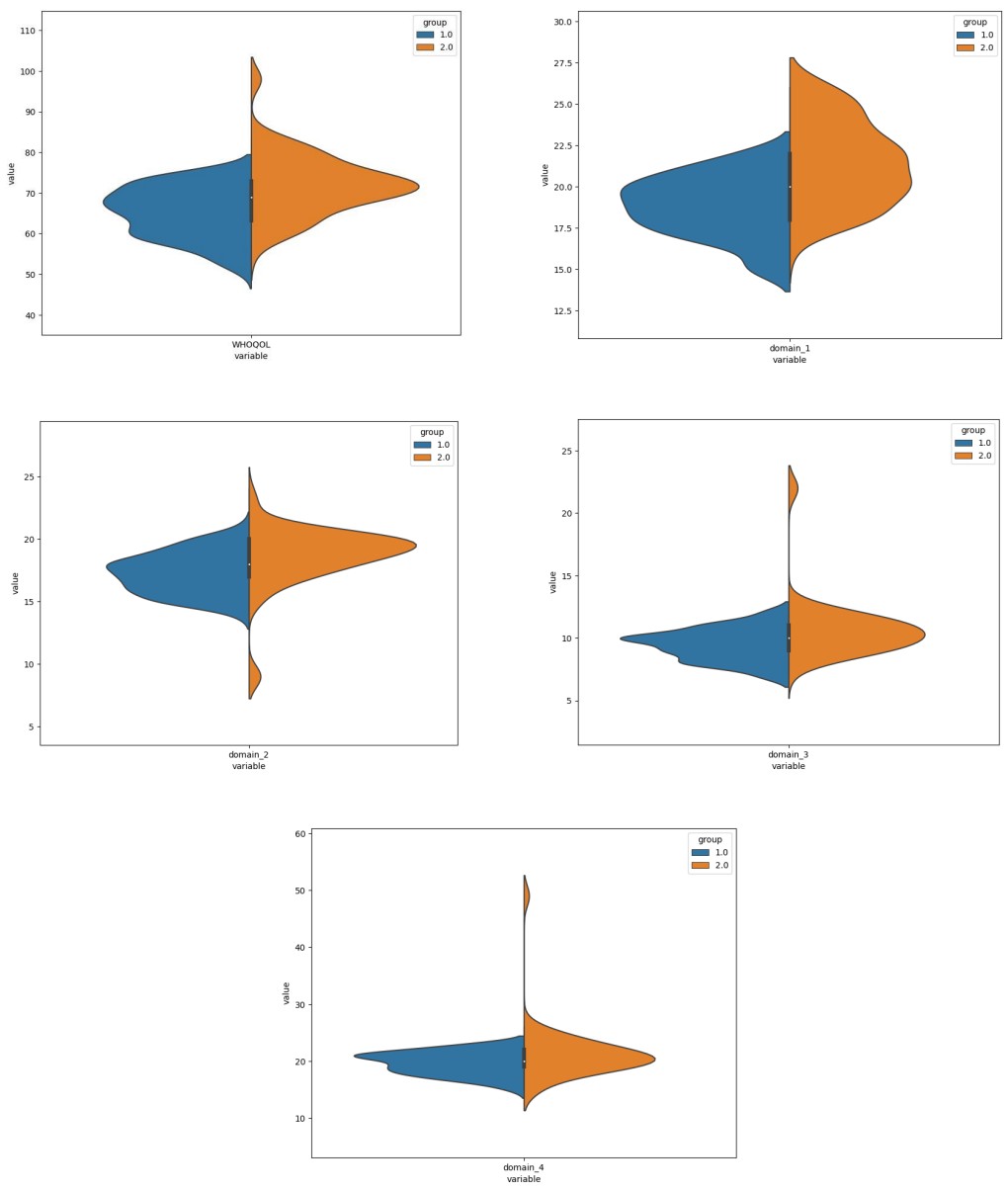

**Figure 3  The violin plots of WHOQoL and each domains.**

is moving (*Crane & Demer, 1997*). For condition 3, the participant was standing on foam with eyes open, so the proprioceptive was removed and the contributions of the visual and vestibular systems was made manifest. The participants with VD implicate the visual system by giving its preference over vestibular feedback for balance. Condition 4 of mCTSIB was standing on foam with eyes closed. In this case, the visual and proprioceptive functions are disturbed and shifting reliance on the vestibular function to maintain balance. People with VD find it difficult to maintain balance in this condition since it relies primarily on the vestibular input. Previous studies demonstrated that standing on foam with eyes closed

was more challenging than standing on the floor, that subjects display greater postural sway (*Goble et al., 2019*). Also, another study reported that VD independently increased the risk of falling two-fold in DM patients assessed by the modified Romberg test. Most of participants failed in the condition 4 to determine vestibular function exclusively. The test instructs the subjects have to maintain balance on a foam-padded surface (to eliminate proprioceptive input) with their eyes closed (to block visual input) (*Agrawal et al., 2010*). Although DPN is the common impairment occurred in T2DM, VD impairment can occur in T2DM patients before they suffer with the DPN (*Deshpande, Hewston & Aldred, 2017*). Failure in conditions 3 and 4 were also suggested to be associated with decreased lower limb muscle strength as well as the poor balance confidence (*Vongsirinavarat et al., 2020*).

The results of dynamic balance tests indicated that T2DM with VD had lower dynamic balance performances measured by TUG and FGA, compared with the group without VD. The patients with high risk of falling recorded highly classified by TUG and FGA categories. The TUG evaluates balance during the walking and turning 180 degrees as well as transfers to and from a seated position (*Podsiadlo & Richardson, 1991*) which was more problematic and increased the risk for falls while turning for patients with VD. Also, the task of walking with head movements in FGA puts even more of a challenge on the vestibular system. A previous study stated that rotating the head to the right, left, up and down during ambulation was the most the difficult tasks of the FGA for subjects with VD (*Shumway-Cook & Woollacott, 2010*). The increased input from the cervical afferents that occurs while rotating the head during walking might conflict with the abnormal information being received in the vestibular nucleus, resulting from VD. Therefore, disruption of the sense of positions in space would occur (*Cohen, 1961*). The results might also imply that the FGA is a more sensitive tool compared with TUG to detect balance problems in persons with VD. Notably, the participants who considered high risk of falling were higher by FGA test than TUG test in two groups.

Furthermore, the mean of muscle strength in lower limb was lower in both groups especially the group with VD. The association between DM and muscle strength was explicit (*Hong et al., 2017*; *Vongsirinavarat et al., 2020*). The greater number of peripheral sensory impairments contributed to reducing muscle strength in ankle and knee movements, as well as significantly reduced mobility and function (*Kraiwong et al., 2019*). More than 80% of participants in this study also had cognitive impairment. In patients with DM, the neurophysiological changes occur due to hyperglycemia and the cognitive decline is evident, notably over 4–6 years after DM diagnosis (*Jia et al., 2020*). A previous study also reported that declined cognitive function could change the gait, imbalance and made the patients became the fallers (*Taylor et al., 2013*).

Regarding the FE, the T2DM patients with VD presented lower score of ABC-16 than without VD. The decreased FE was also proposed to be potentially more debilitating than a fall (*Hewston & Deshpande, 2018*). Previous studies found that neuropathic pain was a strong factor increased fear of falling in T2DM patients and obviously hamper the activities (*Brod et al., 2015*). In patients VD, mostly FE comes because of dizziness and feeling unstable when standing or doing some activities. Previous studies justified the

effectiveness of gaze and balance exercises to improve balance and confidence respectively with moderate effect size (*Aldawsary & Almarwani, 2023*; *Bhardwaj & Vats, 2014*).

The participants with VD in this study were presented with greater limitations of social participation monitored by PSSS-Ina and LSA-Ina. The proportion of participants who were considered having severe restricted participation was also significantly greater. The symptoms associated VD specifically dizziness appeared to increase risk of imbalance and falls (*Agrawal et al., 2010*) and effect on functioning, especially on daily activities and social participation (*Komalasari et al., 2022*). Dizziness is also a condition inducing emotional stress increases isolation, reduces self-autonomy, and self-control as well worsening FE that might highlighted an increased in restriction and disability (*Weidt et al., 2014*). The patients tended to avoid activities, environments and situations that provoke the symptom, and required assistance to stand and ambulate, therefore making them more dependent (*Lasisi & Gureje, 2010*). Vestibular rehabilitation shows positive impact to reduce dizziness feeling and improve QoL significantly (*Giray et al., 2009*; *Ribeiro et al., 2017*)

Obviously, T2DM patients with VD showed significantly lower QoL presented by total score and the scores of all domains of WHOQoL. A previous study explained that DM patients had significantly reduced QoL due to many factors resulting from micro and macrovascular complications. The decreased balance control due to the deterioration of sensory, cognitive and musculoskeletal systems consequently impaired QoL (*Dunsky, 2019*). QoL was also consistently associated with a higher prevalence of disability in all states, as well as with a progression of disability states (*Bourdel-Marchasson, Faqot-Campagna & J, 2007*).

This study had some limitations. The VD condition was identified using clinical tests, *i.e.,* the HIT, the DHT and the SRT. To confirm VD, at least one positive VD tests was used. The goggle glasses were not used to confirm the eyes saccades or nystagmus for all tests. Further studies might differentiate the types of VD and find the association with the balance performance, falls-efficacy, social participation and QoL. Also, the exploration in people without T2DM but having VD would add some confirmation of our findings. In addition, the contribution of psychological factors should add a more holistic approach to the issue of participation and QoL. A cross-sectional design like this study would show the occurrences of each factors that might be associated with the limitations of social engagement as well as the reduced QoL in participants with and without VD. Further studies exploring the contributions as well as the management which could prevent or improve the deterioration are warranted.

## CONCLUSION

The ability to maintain balance declined, balance confidence and physical performances were found to reduced accompanied by the declined social participations and QoL in T2DM with VD compared to the group without VD. The VD complications need to be addressed in clinical settings. The comprehensive screening and management of vestibular dysfunction should be applied. The specific rehabilitation program of vestibular rehabilitation, balance and physical training should be encouraged to improve mobility and balance confidence as well as social participation.

## ACKNOWLEDGEMENTS

The authors would like to express the gratitude to all participants in the study and personnel from the clinical settings used for data collection.

### Funding

The authors received no funding for this work.

### Competing Interests

The authors declare there are no competing interests.

### Author Contributions

- Dwi Rosella Komalasari conceived and designed the experiments, performed the experiments, analyzed the data, prepared figures and/or tables, authored or reviewed drafts of the article, and approved the final draft.
- Mantana Vongsirinavarat conceived and designed the experiments, performed the experiments, analyzed the data, prepared figures and/or tables, authored or reviewed drafts of the article, and approved the final draft.
- Vimonwan Hiengkaew conceived and designed the experiments, performed the experiments, analyzed the data, prepared figures and/or tables, authored or reviewed drafts of the article, and approved the final draft.
- Nantinee Nualnim conceived and designed the experiments, performed the experiments, analyzed the data, prepared figures and/or tables, authored or reviewed drafts of the article, and approved the final draft.

### Human Ethics

The following information was supplied relating to ethical approvals (i.e., approving body and any reference numbers):

This study was a cross-sectional, comparative research. It was conducted after the approval from the ethics committee of Mahidol University Institutional Review Board (MU-CIRB 2020/098.2004) and the Medical Faculty, Universitas Muhammadiyah Surakarta, Indonesia (No. 3009/B.1/KEPK-FKUMS/VII/2020).

### Data Availability

The data is available at NCBI: PRJNA1070244.

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
