# Peer review of "Balance performance, falls-efficacy and social participation in patients with type 2 diabetes mellitus with and without vestibular dysfunction"

_PeerJ, doi:10.7717/peerj.17287_

## Round 0.1 · original submission · Major Revisions

Dear Authors

We have received the evaluation of your manuscript from two specialized reviewers. The reviewers have found several limitations and questions about your manuscript that need to be addressed so that your study can be re-reviewed and considered for publication.

In the current format the manuscript cannot be published, the reviewers suggest major changes that need to be addressed one by one. Some of the issues that need to be addressed are related to the introduction, there are also several clarifications regarding material and methods and results that need to be better specified.

I believe that the issues suggested by the reviewers are appropriate and your manuscript can benefit from all the recommendations.

Reviewer 1 ·

Basic reporting

Abstract:
In the results section, the acronyms TUG and FGA should be changed for their expanded term, as they are not previously mentioned in the abstract.

Introduction:
Line 128: Please provide more specified information regarding the outcome measure extracted related to DPN lower extremity pain. As it was assessed with Numeric Rating Scale, I would suppose that intensity of pain was evaluated. Additional information regarding how it was asked would be needed, for example, current general intensity of pain, in the affected/more affected lower extremity? Overall pain intensity in the last week?

Experimental design

Materials and methods:
2.1 Participants:
Line 98: It should be stated the justification for including only T2DM patients older than 40 years, and also why having the diagnosis of T2DM after at least 5 years.
Lines 99-100: ¿Were participants with corrected visual impairment (use of eyeglasses) considered for inclusion?
Lines 105-110: It should be indicated why authors expected a prevalence of 22%. Was this based on the results of other studies, or a pilot study of this research, etc.?
Additionally, authors should mention the software to carry out the mentioned sample size calculations.

2.2 Procedure:
Lines 112-133: Diagnostic and screening procedures are properly described and justified.
Lines 135-136: Please change the position of the acronym “(mCTSIB)” to the end of the sentence, so it would be displayed as “Modified Clinical Test of Sensory Interaction of Balance (mCTSIB)”
Lines 146-150: Timed up and Go test is a widely used test for assessing multiple related functional outcomes. Please provide a proper justification that ensures timed up and go test as a valid tool for assessing dynamic balance. Functional Gait Assessment, and the other sections of “Self-efficacy” and “Social participation” are properly justified.

Validity of the findings

Results:
Lines 188-189: Please provide the reference for setting the cut-off point of >100 mg/dL for high fasting glycemic levels.

Table 1: Please provide the numeric data (mean and SD) for the total blood pressure, body mass index and number of following within groups.

Table 2: I could not find in the text how the “Number of impaired sensory system(s)” was explored. Please provide this information.

Table 3: I find very appropriate not only analysing the quantitative values regarding continuous data, but additionally providing differences between groups through cut-off points for clusterization. Nevertheless, the cut-off points should be representative of the sample recruited in the study. I find difficulties for extrapolating the cut-off point of FGA (<22) (line 244) from the referenced study to the population included in the present study. Age ranges must be different as in the mentioned study (60-90 years) and belonging to a community-dwelling population. The 22/30 point cut-off may not be a reliable measure for clusterizing patients in your target population. Please, find an appropriate cut-off point for the mentioned variable representative of your population, or eliminate the cluster.
This same observation applies to the Cut-off points for ABC-16, and FTSTS, as the population included in the studies referenced differs from the population included in the present study (inclusion.
In any case, I think it would alse be interesting to provide the age range of the included population anyway.

Please, provide a standard acronym for Five Times Sit to Stand Test, as it is sometimes referenced as FTSTS (lines 149 and 234) and FTSST (line 205 and table 2).

Additional comments

General observation for results and discussion:
It is appropriate that authors have concluded the difference in the results based on p values, nevertheless, there are some results that seam not to differ greatly between groups, such us TUG, ABC-16 or PSSS-Ina (Table 3). I think extracting the effect size of the difference between groups may add great value and a critical point of view for the differences observed. Additionally, it would be of interest to contrast these results with previously reported minimal detectable changes, and minimally clinically relevant changes for the differences observed, as they may nor may not exceed these values, adding great and useful information for clinicians in order to know the impact of vestibular dysfunction in balance outcomes in T2DM patients.

Reviewer 2 ·

Basic reporting

This paper presents a comparative study of individuals with Type 2 Diabetes Mellitus (T2DM) with and without vestibular dysfunction (VD). The paper is well-written, with concise sentences, a comprehensive introduction that provides background information, ample reference support, and a detailed description of the methods employed.

However, there are some areas that may benefit from clarification:
1. In the introduction, the authors introduce three systems that affect postural control, followed by two microvascular complications of T2DM that affect postural control, and then introduce the vestibular system. This may be confusing for readers. It may be helpful to emphasize that diabetic peripheral neuropathy (DPN) and diabetic retinopathy (DR) are associated with visual and proprioceptive systems, respectively.

2. The introduction could benefit from a discussion on the importance of studying the association between VD and balance performance, falls-efficacy, social participation, and quality of life. This would help contextualize the study and highlight its potential implications for clinical practice.

Experimental design

The method used in this study is rigorous because the researchers excluded participants with visual and proprioception disabilities, as well as those with other diseases. Additionally, the confirmation of visual disability was validated using three tests.

However, it would be helpful if the authors provided more detailed information on the t-test used, including whether it was one-tailed or two-tailed.

Furthermore, instead of using a table to present their findings, a violin plot may be a more effective way to convey their conclusions.

Validity of the findings

Another point of confusion is that given the known association between vestibular dysfunction and balance performance, falls-efficacy, and social participation, it is unclear why it is necessary to study and compared individuals with T2DM with and without vestibular dysfunction. It may be more appropriate to include two separate groups in the study: one consisting of healthy individuals and another comprising individuals with vestibular dysfunction but without T2DM. This would help to support the conclusions presented in the paper.

---

## Round 0.2 · Minor Revisions

Dear authors,

Please, we encourage you to review again reviewer 1´s minor revisions.

Regards,

Reviewer 1 ·

Basic reporting

Abstract: Revisar por qué quitaron el TUG
Introduction: Properly corrected and edited.

Experimental design

Methods:
Lines 119-120: Please, authors are encouraged to provide the software for performing the sample size calculations as suggested in the previous review process.
Line 128: Please, expand the “Personal and clinical characteristics”, and mention the variables assessed in Table 1 (age, sex instead of gender, blood pressure, BMI, and FGL). Mention the cut-off points for blood-pressure (properly justified), and for BMI. There might be an error in the cut-off point for blood pressure categories in Table 1, please, review the hypertension cut-off if it would be ≥140/90 mmHg or >140/90 mmHg.
Line 176-177: Please, provide the correlation coefficient for the number “0.84”, i.e., Pearson’s r, Spearman’s rho, etc.
Line 211: The cut-off point of “13” may be incorrect. Please review the study from Komalasari et al., 2022, as I identified a “31” point cut-off for severe participation restriction.

Validity of the findings

Results:
Line 231 and Table 1: Please, review the data concerning the categories of DM duration (years), as in Table 1, readers may get confused with the overlap between the 5-10 year category, and ≥10. Modify the data as you consider more suitable, such as 5-9 and ≥10, or 5-10 and ≥11.
Line 233 and Table 1: Please, correct the percentage of participants with hyperclicemia in both T2DM with VD, and T2DM without VD groups, as they were not actualized.
Table 1: Modify the term “Gender” for “Sex”. Change the term Hypertension, for “Blood pressure” or similar. Maintian the terms “Systole” and “Diastole” or change them for “Systolic” and “Diastolic” respectively. Modify the terms “Yes” for “Hypertension” and “No” for “Normal”. Please, review the cut-off points for blood pressure as abovementioned. Add the coefficient of BMI next to that term (kg/m2), and further include the cut-off points for normal, overweight and obesity next to them. Please, unify the formatting of the identitation and categories, I recommend eliminating the numbers, such as “(1)”, “(2)” along all the categories.
Please provide the chi-squared p-value, for all the categorical variables. I recommend providing the p-value paced on the row of the first category. Therefore, the p-value of the Gender should be downgraded 1 row, and the Numer of falling upgraded 1 row. Please verify the p-value on the Hypertension, systole and diastole, they all may have been upgraded 1 row, as ther is no p-value on the category of “yes” / “no” (hypertension). Please, review this information.

Table 2: Please provide the cut-off points for visual acuity [i.e. Normal (≥75 points/letters…)]. Please provide the chi-squared p-value for the numer of impaired sensory systems, and across the Moca-Ina categories. Additionally, include the cut-off points for MoCA categories and for PSSS-Ind

Additional comments

Congratulations for the authors for modifying the previously requested suggestions, the quality of the manuscript is being greatly enhanced.

Reviewer 2 ·

Basic reporting

The revisions addressed all comments.

Experimental design

The authors applied t-test to address the comments and looks great to me.

Validity of the findings

The authors represent the comments as the limitation of this study, which is acceptable.

---

## Round 0.3 · accepted · Accept

Dear Dr. Komalasari,

Thank you for your submission to PeerJ.

I am writing to inform you that your manuscript - Balance performance, falls-efficacy and social participation in patients with type 2 diabetes mellitus with and without vestibular dysfunction - has been Accepted for publication. Congratulations!